# Nanomaterial Exposure, Extracellular Vesicle Biogenesis and Adverse Cellular Outcomes: A Scoping Review

**DOI:** 10.3390/nano12071231

**Published:** 2022-04-06

**Authors:** Thais S. M. Lima, Wanderson Souza, Luths R. O. Geaquinto, Priscila L. Sanches, Ewa. L. Stepień, João Meneses, Eli Fernández-de Gortari, Nicole Meisner-Kober, Martin Himly, José M. Granjeiro, Ana R. Ribeiro

**Affiliations:** 1Directory of Life Sciences Applied Metrology, National Institute of Metrology Quality and Technology, Rio de Janeiro 25250-020, Brazil; thaiismellolima@gmail.com (T.S.M.L.); wstudzel@gmail.com (W.S.); luthsgeaquinto@gmail.com (L.R.O.G.); plaviolasanches@gmail.com (P.L.S.); 2Postgraduate Program in Biotechnology, National Institute of Metrology Quality and Technology, Rio de Janeiro 25250-020, Brazil; 3Postgraduate Program in Translational Biomedicine, University Grande Rio, Duque de Caxias 25071-202, Brazil; 4Faculty of Physics, Astronomy, and Applied Computer Science, Jagiellonian University, 30-348 Kraków, Poland; e.stepien@uj.edu.pl; 5NanoSafety Group, International Iberian Nanotechnology Laboratory, 4715-330 Braga, Portugal; joao.meneses@inl.int (J.M.); eli.fernandez@inl.int (E.F.-d.G.); 6Department of Biosciences & Medical Biology, University of Salzburg, 5020 Salzburg, Austria; nicole.meisner-kober@plus.ac.at (N.M.-K.); martin.himly@plus.ac.at (M.H.); 7Dental School, Fluminense Federal University, Niterói 24020-140, Brazil

**Keywords:** nanomaterials, nanotoxicology, extracellular vesicles, cell communication

## Abstract

The progressively increasing use of nanomaterials (NMs) has awakened issues related to nanosafety and its potential toxic effects on human health. Emerging studies suggest that NMs alter cell communication by reshaping and altering the secretion of extracellular vesicles (EVs), leading to dysfunction in recipient cells. However, there is limited understanding of how the physicochemical characteristics of NMs alter the EV content and their consequent physiological functions. Therefore, this review explored the relevance of EVs in the nanotoxicology field. The current state of the art on how EVs are modulated by NM exposure and the possible regulation and modulation of signaling pathways and physiological responses were assessed in detail. This review followed the manual for reviewers produced by The Joanna Brigs Institute for Scoping Reviews and the PRISMA extension for Scoping Reviews (PRISMA-ScR): checklist and explanation. The research question, “Do NMs modulate cellular responses mediated by EVs?” was analyzed following the PECO model (P (Population) = EVs, E (Exposure) = NMs, C (Comparator) = EVs without exposure to NMs, O (Outcome) = Cellular responses/change in EVs) to help methodologically assess the association between exposure and outcome. For each theme in the PECO acronym, keywords were defined, organized, and researched in PubMed, Science Direct, Scopus, Web of Science, EMBASE, and Cochrane databases, up to 30 September 2021. In vitro, in vivo, ex vivo, and clinical studies that analyzed the effect of NMs on EV biogenesis, cargo, and cellular responses were included in the analysis. The methodological quality assessment was conducted using the ToxRTool, ARRIVE guideline, Newcastle Ottawa and the EV-TRACK platform. The search in the referred databases identified 2944 articles. After applying the eligibility criteria and two-step screening, 18 articles were included in the final review. We observed that depending on the concentration and physicochemical characteristics, specific NMs promote a significant increase in EV secretion as well as changes in their cargo, especially regarding the expression of proteins and miRNAs, which, in turn, were involved in biological processes that included cell communication, angiogenesis, and activation of the immune response, etc. Although further studies are necessary, this work suggests that molecular investigations on EVs induced by NM exposure may become a potential tool for toxicological studies since they are widely accessible biomarkers that may form a bridge between NM exposure and the cellular response and pathological outcome.

## 1. Introduction

Nanomaterials (NMs) impact our daily lives due to their numerous applications, including chemistry, space, automotive, nutrition, electronics, cosmetics, textiles, medical devices, and pharmaceutical products, among others [1]. They exhibit exciting and novel physicochemical properties that differ from the intrinsic structural properties exhibited by their bulk equivalents [1,2,3,4]. Even though NMs offer multiple technological advantages, understanding their physicochemical properties and their interactions with the biological environment are critical elements for hazard identification, in combination with the knowledge gained on exposure scenarios necessary for risk assessment [4]. Growing evidence demonstrates that NMs are easily accumulated and are difficult to eliminate by the human body, raising concerns about their potential harmful health effects [5]. The most common routes of NM exposure to humans are through inhalation, ingestion, dermal exposure, and medical devices [6]. A plethora of studies have already revealed toxicity associated with NMs, attracting the attention of various interested entities [4,6]. The literature data reveal that inflammatory stimuli together with cytokine overproduction, increased reactive oxygen (ROS), and nitrogen (RNS) species production, are referred to as the primary NM-induced toxic effects, in route to any of the apoptosis, necrosis, and autophagy-mediated cell-death mechanisms that ultimately lead to cytotoxicity [6,7]. 

Biological processes such as cell growth, differentiation, and response to different internal and external stimuli are coordinated by intracellular metabolic pathways. However, external stressors (e.g., NMs) can cause failure in this communication, disturbing homeostasis [5,8,9,10,11]. Intercellular communication occurs in different ways; however, extracellular vesicles (EVs) identified in various biological fluids (saliva, blood, urine, milk, and sputum fluid) have been found as novel key mediators. Due to their own biologically active cargo, EVs cells can exchange molecular messengers over a distance [11,12,13]. Cells can secrete different types of EVs such as exosomes, microvesicles (MVs) or microparticles, and apoptotic bodies, which are classified according to their route of biogenesis and partially differ in size as well as cargo [14,15]. Recently, EVs have been receiving increasing attention as signal transducers because of their bioactive cargo (including RNA, proteins, lipids, and metabolites as well as DNA) that induce changes in the recipient cells’ physiology [11,16,17,18]. EVs are involved in several biological and pathological processes, including angio- and tumor genesis, metastasis, inflammation and regeneration apoptosis, aging, and autoimmune diseases.

Furthermore, the microenvironment of the producing cell can dynamically modulate the EV biogenesis, thereby altering its cargo and functions, which provides additional external cellular factors that modulate cell communication. The inhibition or stimulation of EV secretion in response to external stimuli can cause alterations in cell communication and result in substantial implications to cell/tissue dysfunction and ultimately pathological consequences [14,17,19]. Emerging evidence suggests that the NM exposure of cells and organs stimulates the secretion and disturbs EV biogenesis by activating various biological processes that include immune-system activation, toxicity reduction mediated by exocytosis of NMs through EVs, and the induction of pro-thrombotic effects in cardiovascular and metabolic diseases [20,21,22,23,24,25]. Interestingly, EVs have demonstrated the importance of explaining the possible outcomes of many physiological processes, including the cytotoxic and inflammatory processes induced by NMs [10,20,26]. With the vast diversity of NMs that humans are widely exposed to, significant roles for EVs in the context of NM toxicology and hence diverse physiological and pathological outcomes can be expected. Therefore, this scoping review (ScR) aims to systematically map the data available in the literature regarding the modulation of cellular responses mediated by EVs after exposure to NMs and the corresponding adverse cellular outcomes.

## 2. Materials and Methods

### 2.1. Theoretical and Methodological Framework

This ScR was conducted according to the PRISMA extension for ScR (PRISMA-ScR) and Joanna Briggs Institute Reviewers’ Manual (see Appendix A for a complete checklist) [27,28,29]. This framework comprises five steps: identifying the research question; identifying relevant studies; studies selection; extracting, charting, and collating the data; summarizing and reporting the results [27].

### 2.2. Focused Question

The research question was based on the PECO strategy (Table 1) [30]. Thus, the research question that guided this ScR was: “Do NMs modulate cellular responses mediated by EVs?”.

### 2.3. Search Strategy

The MEDLINE/PubMed, Science Direct, Web of Science, Scopus, Cochrane, and EMBASE databases were comprehensively consulted using the search strategies generated using the following terms: “extracellular vesicles,” “nanomaterials,” “cellular responses,” “modulations of EVs” and their synonyms. The search algorithms were formatted for compatibility with each database. Article search was carried out on 28 April 2020, 7 May 2020, 29 September 2020 and updated on 30 September 2021. In addition to those searches, a comprehensive search of gray literature was conducted on 29 September 2020. Detailed research strategies are provided in Appendix A (see Appendix A). Filters were not used to exclude articles regarding language, type of articles, and publication dates. In each database, export files were generated and uploaded to the systematic review software online Rayyan QCRI [31]. Duplicate articles with at least 90% similarity automatically identified by the software were analyzed and manually excluded by the author TL.

### 2.4. Eligibility Criteria

The primary inclusion criteria for this review were the exposure to any NMs, biological models, and outcomes described in the literature that must have fallen within at least one of the specified outcome groupings of cellular response induced by EVs, such as cargo, number, biophysical alterations in EVs such as inflammation, coagulation, immunologic activation, referred to as ‘modulations in EVs’. Possible comparator group(s) included a control group not exposed to NMs. Therefore, in vitro, in vivo, and clinical trials that analyzed EV modulations induced by exposure to NMs were included. Studies that did not isolate EVs derived from cells without exposure to NMs were excluded. Only studies with original data were included in this ScR. Therefore, for example, review studies, reviews and book chapters were excluded. The complete list of eligibility criteria can be seen in Appendix A (see Appendix A).

### 2.5. Article Selection, Screening Process, and Data Extraction

Before selecting articles, a calibration test involving a set of 100 articles was performed to refine the selection process and ensure a high level of inter-examiner correlation. Then, we calculated the index agreement (Kappa coefficient) between the four researchers (L.R.O.G., P.L.S., T.S.M.L. and W.S.). Articles were selected in two stages. In the first step, the selection of studies was performed by screening titles and abstracts. The next step was screening full-text articles based on relevance to the theme. Both steps were conducted in fulfillment of the inclusion criteria by four independent reviewers who were blind to each other (L.R.O.G., P.L.S., T.S.M.L. and W.S.). The disagreements between the reviewing authors were solved through careful discussion, and any remaining disagreements were solved by a fifth reviewer (A.R.R.). The PRISMA flow diagram (Figure 1) demonstrates the selection process and indicates the number of articles excluded at each screening phase. Complete data from studies included in the final review, such as type of NMs and EVs, isolation and characterization of EVs and their respective outcomes, are presented in detail in the supplementary information (see Appendix A).

### 2.6. Assessment of Reliability

In the process of critical analysis and the attribution of more transparent and harmonized reliability categories, four tools were used: ToxRTool [32], ARRIVE Guideline 2.0 [33], Newcastle-Ottawa Scale (NOS) [34], and MISEV [35]. The ToxRTool tool classified and categorized toxicological studies by scoring criteria in different reliability categories, being 1: reliable without restrictions, 2: reliable with restrictions, 3: unreliable, and 4: not assignable [32]. The Newcastle-Ottawa Scale (NOS) assessed the quality of non-randomized studies of the case-control type. With this tool, the study is judged from three broad perspectives: the selection of study groups (0–4 points), the comparability of groups (0–2 points), and the verification of exposure (0–3 points) [34]. The ARRIVE Guideline 2.0 checklist was used to assess data reliability presented in studies involving animals. The degree of compliance of the studies with ten essential items included in the studies report was analyzed, namely 1. study design; 2. sample size; 3. inclusion and exclusion criteria; 4. randomization; 5. blinding; 6. outcomes; 7. statistical methods; 8. experimental animals; 9. experimental procedures and 10. results [33]. The minimum information guideline for studies of EVs published by MISEV describes the techniques used for a basic characterization of EVs [35]. Thus, we assessed whether the included studies followed this guideline, and whether there was registration on the EV-TRACK platform and EV-METRIC calculation. Key sources are recommended to be followed before submitting the manuscript for peer review [31,35]. Discrepancies between reviewers were resolved by consensus, and, in case of persistent disagreement, the assessment was made by a senior reviewer (J.M.G.). Complete data are presented in supplementary information tables (see Appendix A).

### 2.7. Bibliometric Data Analisys

VOSviewer (software version 1.6.17, Leiden, The Netherlands) was used to systematically map and visualize the available data in the literature [36]. The first step was to generate a bibliographic-citation (.ris) file from the systematic-review software. Such files contained information about the 2944 articles, and were used as inputs to VOSviewer. Afterwards, the type and unit of analysis were defined as co-occurrence and keywords, respectively. The co-occurrence threshold, i.e., the minimum number of keyword occurrences in all articles, was set as 25, and 128 were obtained. The next step involved this ScR author’s expertise and regarded the manual selection of the final keywords. By doing so, it was possible to decrease the complexity of the network and ensure the credibility of the process. Hence, 40 of the 128 possible keywords were selected. The last step involved setting the analytical parameters, such as modularity-based clustering [37] and association-strength normalization, as default. In short, a keyword co-occurrence network representative of the 2944 articles was obtained, with 40 keywords and 654 connections. Each connection represents an article in which two keywords occur together. 

The ultimate goal of the bibliometric data analysis was to map and visualize the 18 articles that met all the eligibility criteria in order to eventually establish relationships between the NM exposure and the biological outcomes. The first step was to transpose the previous network to GEPHI (software version 0.9.2, Menlo Park, CA, USA), which allowed the deconstruction of such a complex network [38]. A geography-markup-language (.gml) file was generated from VOSviewer and used as the input to GEPHI. The next step involved several visual changes, such as removing the directionality of the connections between the keywords, which was added by default, and selecting the 300 stronger connections, which allowed a better understanding of the network. Afterwards, the ScR author’s expertise was, once again, needed. It regarded the manual selection of the most representative keywords of the 18 articles among the 40 possibilities. This selection involved the comparison of the 40 keywords with each set of keywords of the 18 articles. Then, a keyword co-occurrence network representative of the 18 articles was obtained, with 15 keywords and 73 connections. All documents used in this subsection are available in Appendix A.

### 2.8. Tabular Data Analysis

To summarize the effect of the physicochemical characteristics of NMs on EV secretion and the consequent biological outcomes from the 18 articles that met all the eligibility criteria, this ScR presents Table 2 and Table 3. As a complement, a visual representation was obtained by GEPHI (software version 0.9.2) [38]. The first step was to define the representative entities, namely author, NM type, NM size, biological origin, fluid collection, EV nomenclature, EV size, EV-enriched markers, EV non-enriched markers, and biological outcomes. The next step was to detail sub-entities per each entity. For instance, each article’s first author was used as a sub-entity of the author entity, and so on. Then, 17 types of connections were set. A couple of examples are author–NM type, author–NM size, NM type–NM size, author–biological origin, author–fluid collection, and biological origin–fluid collection, among others. The last step regarded using the following algorithms: Force Atlas 2 (with all behavior-alternative fields checked, over 1 min), Expansion, Label Adjustment, and Noverlap. The first algorithm was used to spatialize the network, while the others fulfilled aesthetic concerns. Overall, a representative network of the 18 analyzed articles was obtained with 10 entities, 110 subentities, 17 types of connections, and 363 connections. All documents used in this subsection are available in Appendix A.

## 3. Results

The bibliographic search identified 4203 studies, including 816 titles from MEDLINE/PubMed, 1085 from Science Direct, 1128 from Scopus, 697 from Web of Science, 14 from Cochrane, and 463 from EMBASE. The gray literature search or cross-references did not identify any relevant study. The 100-sample analysis studies used to calculate the coefficient of agreement (Kappa − κ = 1) indicated a perfect level of agreement among the four researchers. We returned with 2944 unique results (1259 duplicates were deleted), of which the titles and abstracts were analyzed. As shown in Figure 1, 2823 studies were excluded since they did not meet the experimental eligibility criteria. In the full-text-screening step for the 121 studies analyzed, only 18 met all the eligibility criteria. They were then included in the final review, following the data-extraction and qualitative-synthesis steps, Figure 1A. The significant discrepancy between the number of identified articles in the initial search and those included in this ScR is because most of the articles that address the topic “EVs” did not expose cells and tissues to NMs. In short, most of the excluded articles addressed EVs as new diagnostic and therapeutic tools, where exposure to NMs and the consequent possible effect on EV modulations was neglected.

### 3.1. Characteristics of the Studies

The studies included in this ScR were published in the last nine years (2012–2021), with 2021 being the year with the highest number of publications, showing that the potential application of EVs in nanotoxicology is still in its infancy. To illustrate the current research hotspots, the bibliometric data were analyzed. Figure 1B,C show the map of keyword co-occurrence and its interconnections among the 2944 unique results and the 18 ultimately selected articles. Each circle represents a keyword, and each color represents a cluster suggesting a similar topic among the publications. The circle size is directly proportional to the keyword occurrence, i.e., to the number of articles in which a keyword occurs [36]. The relation between keywords is represented by a line whose thickness is directly proportional to the number of articles in which two keywords occur together [36]. On one side, the network that illustrates the 2944 articles (Figure 1B) presents a higher degree of complexity, with 40 keywords subdivided into 4 clusters and 300 connections. On the other side, the network that depicts the 18 articles (Figure 1C) contains 15 keywords subdivided into 4 clusters and 73 connections. From Figure 1B,C, each keyword kept its cluster association. Such behavior was expected since Figure 1C was built by deconstructing the initial network, Figure 1B. Overall, both networks suggest that NMs exposure affects EVs biogenesis, miRNA cargo, and protein expression. All of these terms are widely involved in cell communication. Moreover, both networks show that most studies involved animal and human models. The main biological activities referred to in the 18 articles were angiogenesis, biogenesis, cell communication, inflammation, and oxidative stress. Among these, biogenesis has the highest frequency, and its correlation with the EV is the strongest.

### 3.2. Biological Models, Exposure to NMs and Correlation with Physicochemical Characteristics

Table 2 summarizes the information regarding the biological models and experimental design included in this ScR and the physicochemical characteristics of NMs and their impact on cytotoxicity (more information about the experimental model and the NMs is presented in the Appendix A). Observed biological impacts ranged from impact on cell viability to modulation of immune activity and function including type 1 immune polarization. Notably, no clear trends for a specific type of NM or surface functionalization can be deduced from the limited number of studies that qualified for our ScR (focusing on NMs and EVs). On the biological impact of physicochemical properties of NM in general much has been published during the past decades. From the studies that were selected for this ScR, eleven studies used human cell lines as models, five used mouse models, and two used both human and animal experimental models. Thirteen articles performed in vitro studies; in vivo studies were used in two articles, two studies performed hybrid assays (in vitro and in vivo), and only one article performed a clinical trial exploring patient-derived EVs. It is important to note that only two studies performed tests that referred to the quality control of the biological sample, namely cell viability and authenticity [26,39]. Medical tests were used in the clinical-trial study to select the sample group and exclude confounding variables such as blood characteristics that could promote changes in the collected biological sample [24]. Most in vivo studies described the origin, weight, sex, age, and maintenance characteristics of the animals used [40,41]. Finally, all the studies included in this ScR reported the performance of at least one triplicate for each experiment performed.

It is well known that physicochemical properties of NMs change upon contact with different environments due to protein corona formation, which represents the most genuine molecular initiating event in bio-nano sciences. Hence, the experimental readout may be impacted by immune-activating or -modulating contaminants. Another very important point to be stressed in this context is dosing. In vitro assays dealing with NMs need to be well controlled in regard to the well-known deviation between administered vs. cell-delivered dose [42]. However, such types of dose effects were not studied in great detail in the studies that were selected for this ScR. Regarding the physicochemical characteristics of NMs, 78% of studies used nanoparticles of different chemical compositions as test substances (e.g., metallic, polymeric, metallic oxides, inorganic, etc.) [24,40,42,43,44,45,46,47,48,49,50]. The remaining studies used nanotubes, nanoclusters, nanosheets, as well as dendrimers [26,49,50,51]. As for the characteristics of NMs, only six of the studies analyzed and reported the zeta potential, two reported the polydispersity index, four analyzed the presence of contaminants (e.g., endotoxins as relevant for immune effects), and three reported the purity as quality control of the NMs. As for the characterization methods of NMs, the following techniques were mainly used: dynamic light scattering (DLS), transmission electron microscopy (TEM), scanning electron microscopy (SEM), inductively coupled plasma-mass spectrometry (ICP-MS), nanoparticle tracking analysis (NTA), X-ray diffraction (XRD) and energy-dispersive X-ray spectroscopy (EDS). Concerning the route of exposure of NMs, in vitro studies used direct exposure in culture medium.

In contrast, two in vivo trials included the respiratory exposure route (intratracheal instillation). Finally, the observational clinical trial did not use any exposure route, as it was assumed that patients diagnosed with pneumoconiosis had been exposed to toxic substances and NMs daily. Furthermore, the results observed with these patients were compared with results obtained from a series of in vitro assays using two immortalized cell lines exposed to NMs. One of the outcomes observed arising from the exposure of experimental models to NMs was reduced viability and cytotoxicity. Overall, 28% of the studies reported that NMs induced cytotoxic effects after the NM internalization process.

On the other hand, 28% of the studies reported that NMs do not induce changes in viability in the concentrations and times of exposure tested. Curiously, 11% of the studies reported an increase in the viability rate after NMs exposure. However, it is important to note that these effects vary according to the physicochemical characteristics of the NMs, and the biological model applied. In contrast, 22% of studies did not report this information.

**Table 2 nanomaterials-12-01231-t002:** Physicochemical characteristics of NMs, exposure conditions, and biological effects observed in the respective experimental models.

Reference	Nanomaterial					Exposure Conditions	Biological Effects ^4^
	Type ^1^	Size ^2^	Morphology/Crystalline Structure	Purity ^3^	Z Potential		
[20]	PEI-SPION NPs	15 nm	-	-	Low density: 4.5 mV High density: 7.7 mV	Immersion CM 2–7 μg/mL 24 h	In vitro Human HMVECs: PEI-SPION NPs uptake did not impact cell viability
[22]	MIONs	-	-	LPS < 0.25 EU ml^−1^	-	Respiratory exposure (Intratracheal instillation) 20 μg in 50 μL PBS Three times at daily intervals (days 0, 2, and 4).	In vivo Mouse BALB/c: NPs transferred across the pulmonary cell membrane and located in lysosomes
[24]	SiO_2_ NPs Occupational NPs	10–20 nm	-	≈ 99.5%	-	Immersion CM 100 µg/mL 24 h Exposed to occupational inhaling	In vitro: Human THP-1 Clinical: pneumoconiosis patients
[25]	Au NPs	P: 20 nm Water: 20.5 nm (T0 h), 20.2 nm (T24 h), CM: 19.5 nm (T0 h), 19.9 nm (T24 h)	-	-	-	Immersion CM 0.1, 1, 10 e 50 μM 24 h	In vitro Human PBMCs: Internalization of Au NPs in the early endosomes and/or in structures resembling MVB
[26]	SWNCTs	200–1000 nm	Fiber-like	>95%	−44.1 mV (pH 12); −23.2 mv (pH 2)	Immersion CM 10 μg/mL 0–48 h	In vitro Mouse PMQ: ↑ SWCNTs uptake with prolonged exposure time No significant cell death. Alteration in primary macrophage morphology
[39]	TiO_2_ NPs ZnO NPs	P: 21 nm; CM: 28.6 nm P: 10 nm; CM: 16.9 nm	-	LPS: NPs < 50 pg/mL; Culture media < 5 pg/mL)	−12.2 ± 0.25 mV 11.4 ± 0.17 mV	Immersion CM 0.5–100 μg/mL 24 h	In vitro Human PBMC: No cell death MDDC: NPs active uptake. No alteration of surface markers In vitro Human MDDC: ↑ Cell death, ↑ Cas dose-dependent, ↑ DNA fragmentation. No NP uptake and no change in surface-marker expression PBMC: No differences in inactivation or expression of CD69 in T-cell. ↓ CD16 on NK-cells
[40]	Fe_3_O_4_ NPs	100 nm	-	-	-	Immersion CM 400, 200, 100, 50, 25 μg/mL 1, 3, and 5 days	In vitro: Human BMSCs: ↑ Cell viability in the optimal working concentration (50 μg/mL)
[41]	MIONs	P: 43 nm Solution: 43 nm	Cubic	LPS < 0.25 EU ml^−1^)	-	Respiratory exposure (Intratracheal instillation) 20 μg in 50 μL PBS days 15, 17, and 19	In vivo Mouse BALB/c: NPs transferred across the pulmonary cell membrane and located in lysosomes ↑ Th1 polarization ↑ Tc1 OVA-sensitized mouse BALB/c Mouse BALB/c: ↑ Activated Th + Tc ↑ Th1, ↑ Tc1 ↑ IFN-γ, ↑ IL-4 ↑ Inflammation
[42]	CaP	1.84 ± 0.48 μm	Spherical or oval	-	−2.49 mV	Immersion CM 250, 500, 1000, 2000 μg/mL 1–3 days	In vitro Mouse RAW264.7: No alteration in proliferation In vitro Human THP-1: No alteration in proliferation
[43]	Au NPs	5, 20, 80 nm	Spherical	-	AuNPs-5: −22.01 ± 1.81 mV, AuNPs-20: −32.17 ± 2.19, AuNPs-80: −55.21 ± 7.34 mV	Immersion CM 1 μg/mL 24–48 h	In vitro Mouse mESCs: LOEC: 5 μg/mL Non-cytotoxic and does not induce ROS. No interference in self-renewal or pluripotency
[44]	Pt NPs	40–50 nm	Spherical, triangular, oval, and rod-shaped		-	Immersion CM 0, 2.5, 5, 10, 20, and 40 μmol/L 24 h	In vitro Human A549 monolayer culture: ↑Viability and proliferation Morphological signals of autophagy ↑ ROS, ↑ Cas3, ↑ LDH, ↑AchE
[45]	Pd NPs	~20 nm	Spherical	-	-	Immersion CM 5–25 μM 24 h	In vitro Human THP-1 Monolayer culture:
[46]	Fe_3_O_4_ NPs	8, 15, 30 nm	Spherical	8 nm (99.9%), 15–20 nm (99.5%) and 20–30 nm (99.0%)	-	Immersion CM 0, 1, 10 e 100 μM 48 h	In vitro Human iNPCs Cortical spheroids culture: Changes in morphology No effects on cell viability, metabolic activity, neurodegeneration, or oxidative stress
[47]	POSS NPs	3–5 nm	Spherical	-	-	Immersion culture media 0- 600 ppm24 and 48 h	In vitro: Human HUVECs: ↑ Viability, ↑ Migration, ↑ Wound healing, ↑ VEGFR-2, HSP-70, Ang-1, and Ang-2, ↑ miRNA-21 and miRNA-155, ↑ VEGF-A and TGF-β, ↓ miRNA-182
[48]	nHAp	<100 nm	Rod-like	97%	-	Immersion CM 100 ug/mL 24 h, 7, and 14 days	In vitro Mice C57BL/6 VSMCs: ↑ ALP, Runx2, and OPN, ↑ Autophagic organelles, ↓ Lysosomal acidification, No effect on the viability, ↑ calcium deposition
[49]	s-GO	50–500 nm	-	LPS free	−55.9 ± 1.4 mV	Immersion CM 10 μg/mL Six days	In vitro Rats Wistar Astrocytes: No impairment of astrocyte morphology or cell density No effect on viability Did not cytotoxic effect
[50]	PAMAM	G2:3 nm G7: 9 nm	-	-	G2-NH_2_: 19.8 mV; G2-COOH: −21.7 mV; G2-OH: 4.8 mV G7 NH_2_: 30.1 mV; G7-COOH: −19.5 mV; 2.8 mV; G7-OH: 2.8 mV	Immersion CM 1–100 μg/mL 24 h	In vitro Human HUVECs: Low cytotoxicity Moderate g1 arrest of cell cycle G2-NH_2_: ↑ICAM-1(CD54), ↑ Apoptosis G7-NH_2:_ ↑ICAM-1(CD54), ↑PS ↑ Apoptosis ↑ Necrosis ↑ Plasma-membrane blebbing, disintegration, and permeability Moderate g1 arrest of cell cycle
[51]	NCs	Ag NCs: 1.3 nm; Fe_3_O_4_ NCs: 3.5 nm	-	-	-	Immersion CM 10–100 μmol/L 24 h	In vitro Human L02: Induce dose-dependent cytotoxicity ↓ Viability In vitro Human HepG2: Not difference in cytotoxicity

^1^ AuNPs (Gold nanoparticles), AgNO_3_ NCs (Silver NCs), CaP (calcium phosphate particles), COOH-terminated (anionic), Fe_3_O_4_ NCs (Iron oxide NCs), Fe_3_O_4_NPs (Magnetic iron oxide nanoparticles), MIONs (Magnetic iron oxide nanoparticles), NCs (Silver and Iron oxide nanoclusters), nHAp (Nano-hydroxyapatite), NH_2_-terminated (cationic), NPs (Nanoparticles), PAMAM (polyamidoamine dendrimers), PEI-SPION NPs (NPs superparamagnetic iron oxide NPs associated with NPs polyethyleneimine), Pd NPs (Palladium nanoparticles). POSS NPs (Polyhedral oligomeric silsesquioxane nanoparticles), Pt NPs (Platinum nanoparticles), s-GO (Small graphene-oxide nano-flakes), SiO_2_NPs (Silicon dioxide nanoparticles), SWNCTs (acid-oxidized single-walled carbon nanotubes), TiO_2_ NPs (Commercial titanium dioxide), ZnO (Commercial zinc oxide). ^2^ P (Primary), CM (Culture Medium). ^3^ LPS (Endotoxin contamination). ^4^ ↑ ((Increase, induction), ↓ ((Decrease, loss), AChE (Acetylcholinesterase), ALP (Alkaline phosphatase), Ang (angiopoietin), A549 (human lung epithelial adenocarcinoma cancer cells), BMSCs (Bone mesenchymal stem cells), Cas (Caspase), CAT (Catalase), DNA (Deoxyribonucleic acid), ER (Endoplasmic reticulum), FGF (fibroblast growth factor), GSH (Glutathione), GST (Glutathione S-transferases), GPx (Glutathione peroxidase), HepG2 (Human hepatocellular carcinoma), HMVECs (human microvascular endothelial cells), HUVECs (Human umbilical vein endothelial cells), IC50 (Half maximal inhibitory concentration), IL-4 (Interleukin-4), iNPCs (Neural progenitor cell), LDH (Lactate dehydrogenase), LHP (Lipid hydroperoxides), LOEC (Lowest observed effect concentration), L02 (Human embryonic liver cell), RAW264.7 (Macrophage-like), MDA (Malondialdehyde), MDDC (Monocyte-derived dendritic cells), NO (Nitric Oxid), OPN (Osteopontin), OVA (Ovalbumin-sensitized), mESCs (mouse embryonic stem cells), PBMC (Primary human peripheral blood mononuclear cells), PCC (Protein carbonylation content), PMQ (non-activated primary mouse peritoneal macrophages), ROS (Reactive oxygen species), Runx2 (Runt-related transcription factor 2), Tc (T cytotoxic cell), Th (T-helper cell), Th1 (T-helper cell type 1), Th2 (T-helper cell type), IFN-γ (Interferon-gamma), THP-1 (Human leukemia monocyte-like), TRX (Thioredoxin), TGF-β (transforming growth factor beta), VEGF (Vascular endothelial growth factor), VEGFR-2 (VEGF receptors), VEGF-A (VEGFR-2 ligand), VSMCs (primary mice vascular smooth muscle cells).

### 3.3. EVs and Outcomes of NM Exposure

Table 3 summarizes information about the process of isolation and characterization of EVs, including enriched and non-enriched EV markers. In addition, important information such as the origin of EVs and the a priori biological conditions, as well as the main biological outcomes induced by EVs are listed. Twelve of the thirteen in vitro studies isolated EVs from the culture medium of cells exposed to NMs, and one study did not isolate EVs. In vivo studies isolated EVs present in bronchoalveolar lavage fluid (BALF) from mice [22,41]. In turn, the clinical trial studied pneumoconiosis-derived EVs isolated from serum samples of patients, EVs derived from healthy patients, and the supernatant of cells stimulated with the NMs [24]. Most of the studies did not fully describe the conditions of the biological model during collection (culture medium, passage, cell viability, reagents used, etc.). The most widely used method for EV isolation was ultracentrifugation, and the majority of articles included within this review referred to exosomes as the class of isolated extracellular vesicles. No article dealt with isolated apoptotic bodies, and in one article, isolated microvesicles were investigated. It is important to note that the term EV was also part of the nomenclature of isolated vesicles in different studies [20,43,46,50].

In most cases, this term was used to designate a pool of isolated EVs without any purification step. TEM, SEM, NTA, Western blotting (WB), and DLS were the most used techniques to characterize EVs. Only eight studies reported analysis regarding EV size, with studies referring to ‘exosome’ isolation reporting mean diameters ranging from 30–200 nm while EVs generally range from 60–1000 nm. Compared with the control, six studies reported no significant differences in EV size in response to NM exposure [26,39,41,45,49,51], while one study observed a decrease in EV size [46], and one study reported an increase in EV size [25]. Interestingly, the impact of NMs on EV biogenesis is evident. Among the eighteen articles, fourteen studies showed that NMs increased EV secretion, especially for studies specifically referring to exosomes [20,24,26,40,41,42,44,45,46,47,48,49,50,51]. Twelve studies explored EV-enriched biomarkers (with CD81, CD63, CD9, and TSG101 being the predominantly analyzed markers), while three studies analyzed additional non-EV markers (GAPDH, Calnexin, and Calreticulin) [40,43,48]. As for possible modulations in EVs induced by exposure to NMs, the studies reported that there might be an increased release of EVs in terms of numbers, protein and miRNA cargo alterations, increased expression of EV-enriched markers and alterations upon EV transfer to recipient cells (Table 3). The prominent biological modulations induced by EVs derived from cells exposed to NMs included immunomodulation, immune activation, inflammation, angiogenesis, healing, and EVs as mechanisms of NM elimination from the cells and tissues.

**Table 3 nanomaterials-12-01231-t003:** Overview of the effects of NM exposure on EVs secretion and consequent biological outcomes.

Reference	Nanomaterials ^1^	Biological Origin and Fluid Collection ^2^	Isolation and Characterization ^3^	EV Nomenclature and Size ^4^	Ev Enriched and not Enriched Markers ^5^	Biological Outcomes ^6^
[20]	PEI-SPION NPs 15 nm	HMVECs; Cell-culture-conditioned media	HSC, MS; TEM, LSCM	EVs; 100 ± 1000 nm	-	↑ EVs associated with apoptotic cell; Intercellular transfer of NMs through EVs PS+ in MCF7, 4T1 or HMVEC co-cultivated with EVs
[22]	MIONs	Mouse BALF	Centrifugation and ultracentrifugation; TEM, Protein dosage, FC	Exosomes 30–90 nm	-	↑ Exosome biogenesis Exosomes MHCI H-2Kd+, MHCII I-Ad+, and CD80+ secreted are of APC origin. **In vitro**: Exosomes are internalized by AM φ, Raw264.7, and iDCs cells co-cultivated with exosomes. ↑ iDC maturation and secretion of cytokines DC1 and IL-12 exosomes mediated. ↑ Polarizing on T-cell differentiation in Th1 and Tc1 exosomes mediated. ↑ Phagocytic index of AM φ and secretion IL-12, and TNF- α exosomes mediated. **In vivo**: Exosomes were distributed in the liver and spleen of BALB/c mice
[24]	SiO_2_ NPs 10–20 nm Occupational NPs	IMR-90 and THP-1; Cell-culture-conditioned media Venous blood from patients (pneumoconiosis and control)	Ultracentrifugation; TEM, NTA, miRNA Isolation and High-Throughput Sequencing, Immunoblotting RT-qPCR, Immunohistochemistry, FC	Exosomes 30–150 nm	-	↑ Exosome biogenesis; ↓ hsa-let-7a-5p, ↓ hsa-let-7i-5p ↑ WASL expression; ↑ Phagocytosis of NPs ↑ Fibroblast transdifferentiation in IMR-90 fibroblasts co-cultivated with exosomes. ↑ Collagen deposition in IMR-90 fibroblasts co-cultivated with exosomes
[25]	Au NPs 19.9 ± 3.3 nm	PBMCs Cell-culture-conditioned media	Centrifugation and ultracentrifugation. TEM, WB, Bradford protein assay, FC, SP-ICP-MS, NTA.	Exosomes 127.0 ± 3.8 nm	TSG101, CD9, and CD81	↑ Exosome size and refractive index; Au NPs internalized in exosomes. PBMC-derived exosomes eliminate Au NPs
[26]	SWNCTs 200–1000 nm	PMQ; Cell-culture-conditioned media	TEM, SEM, RAMAN	Exosomes; 50–100 nm EVs; 100–400 nm	-	↑ Exosome biogenesis on the surface of macrophages; Exocytosis of SWCNTs through exosomes and EVs; Internalization sustained of SWCNTs in PMQ following exocytosis by exosomes.
[39]	TiO_2_ NPs 28.6 ± 3.2 nm ZnO NPs 16.9 ± 0.3 nm	PBMC and MDDC; Cell-culture-conditioned media	Ultracentrifugation, TEM, Protein concentration, NTA, FC	Exosomes; 30–100 nm	CD81, CD63, CD61, CD86, CD95/FasL MHCI and MHCII	No alterations in exosome secretion, morphology, size, number, or protein cargo
[40]	Fe_3_O_4_, NPs 100 nm	BMSCs Cell-culture-conditioned media	Centrifugation and ultracentrifugation TEM, NTA, WB	Exosomes 116.2 nm	CD9, CD63, CD81, TSG101 Calnexin	↑ Exosome biogenesis. No morphological, size, shape, or electron-density alterations **In vitro**: ↑ Proliferation, migration, and angiogenesis in HUVECs and HSFs co-cultivated with exosomes ↑ miR-21-5p, ↓SPRY2, ↑PI3K/AKT and ERK1/2↑ Exosome biogenesis. No morphological, size, shape, or electron-density alterations **In vitro**: ↑ Proliferation, migration, and angiogenesis in HUVECs and HSFs co-cultivated with exosomes ↑ miR-21-5p, ↓SPRY2, ↑PI3K/AKT and ERK1/2↑ Migration, proliferation, and tube formation in HUVECs co-cultivated with exosomes ↑ Migration in HSFs co-cultivated with exosomes ↑ Mature miR-21-5p, VEGF, HIF-1α, PDGFRα, and bFGF in HUVECs and HSFs co-cultivated with exosomes **In vivo**: ↑ Wound closure, ↑ Density of blood vessels, ↑ Collagen deposition, ↓ Scar widths, ↑ Angiogenesis, ↑ Formation of sebaceous glands and hair follicles exosomes induce
[41]	MIONs 43 ± 5 nm	Mouse BALF	Centrifugation and ultracentrifugation; TEM, Protein dosage, WB, EDS, ICP-MS, FACS	Exosomes 30–90 nm	TSG101	No morphological or size changes of exosomes **In vitro**: ↑ Exosome biogenesis. Exosomes induce iDC maturation. Exosomes induce sensitized T-cell activation and differentiation **In vivo**: ↑ Exosome biogenesis in the alveolar region of BALB/c mice. Exosomes MHCI H-2Kd+, MHCII I-Ad+, CD80+, and CD86- secreted are of APC origin. Exosomes induce a systemic immune response by being eliminated from alveolar spaces
[42]	CaP 1.84 ± 0.48 μm	RAW264.7 and THP-1 Cell-culture-conditioned media	Total Exosomes Isolation Kit EXOCET kit, WB, ICP-OES, DLS	Exosomes: 30.2 ± 8.6 nm Exosome aggregation: 196.3 ± 73.2	CD9, LAMP-1	↑ Exosome biogenesis. No alterations in Ca content
[43]	Au NPs 5, 20, and 80 nm	mESCs; Cell-culture-conditioned media	Ultracentrifugation and filtration TEM, WB, NTA, QCM-D, LC-MS/MS	EVs 60–70 nm	CD63, HSP70, and Flotilina-1 Calreticulin	EVs-5: ↑ The rigidity of EVs, differentially expressed protein profile, and cellular uptake. ↓ Proliferation and migration of 4T1 cells co-cultivated with exosomes. ↓ Cofilin expression and Erk phosphorylation sEV-20 and sEV-80: negligible effects
[44]	Pt NPs 40–50 nm	A549 Cell-culture-conditioned media	Differential centrifugation and ExoQuick; DLS, NTA, TEM, SEM, EXOCET^TM^, FP, qRT-PCR, ELISA, BCA	Exosomes 90–100 nm	TSG101, CD81, CD63, CD9	↑ Exosome biogenesis ↑ Exosome total protein concentration ↑ Concentrations of TSG101, CD9, CD63, and CD81 proteins, Typical morphology and no significant difference in size were observed
[45]	Pd NPs ~20 nm	THP1; Cell-culture-conditioned media	Differential ultracentrifugation and ExoQuick^TM^; DLS, NTA, SEM, TEM, EXOCET^TM^, FP, BCA, qRT-PCR, Enzyme-linked immunosorbent assay, ELISA and WB	Exosomes 50–80 nm	TSG101, CD9, CD63 and CD81	↑ Exosome biogenesis. ↑ Exosome cytokine and chemokine levels (IL-6, MCP-1, IL-8, GM-CSF, TNF-α and IL-1β). ↑ TSG101, CD9, CD63, and CD81 Exosome markers expression levels; No morphological changes were observed
[46]	Fe_3_O_4_ NPs 8, 15, and 30 nm	iNPs; Cortical spheroids Culture-conditioned media	Differential ultracentrifugation and PEG-based method; RT-PCR, NTA, TEM	EVs 200–250 nm	CD63, CD81, Alix, TSG101, Syntenin1, ADAM10, RAB27b, and Syndecan	8 and 15 nm: ↑ EV biogenesis. 30 nm: ↓ EV mean size No morphological changes were observed. Differential gene expression of EV biogenesis markers (CD63, CD81, Alix, TSG101, Syntenin1, ADAM10, RAB27b, and Syndecan) by different size NPs
[47]	POSS NPs 3–5 nm	HUVECs; Cell-culture-conditioned media	Exo-spin™ kit and centrifugation SEM, TEM, FC, AChE activity	Exosomes	CD63	↑ Exosome biogenesis
[48]	nHAp <100 nm	VSMCs; Cell-culture-conditioned media	Centrifugation and ultracentrifugation TEM, DLS, WB, granular analysis	Exosomes 100–133 nm	Alix, TSG101, and CD9 GAPDH	↑ Exosome biogenesis. ↑ Ca content
[49]	s-GO 50–500 nm	Astrocytes; Cell-culture-conditioned media	Centrifugation WB, LSCM, AFM, NTA, FTIR-ATR spectroscopy, UVRR	MVs 50–500 nm	Flotillin-1	↑ MV biogenesis; Altered protein content in EVs; No alteration in EV morphology or size; ↑ PSCs in cortical neurons co-cultivated with MVs ↓ Neuronal stiffness. ↑ Synaptic activity
[50]	PAMAM G2:3 nm G7: 9 nm	HUVECs Cell-culture-conditioned media	Centrifugation NTA, TEM, FC	EVs 120 nm	-	↑ EV biogenesis; ↑ EVs CD105+, PS+, TOM20+
[51]	NCs Ag NCs: 1.3 nm Fe_3_O_4_ NCs: 3.5 nm	HepG2; Cell-culture-conditioned media	Centrifugation, filtration, and ultracentrifugation TEM, DLS, LSCM, EDS, SEM	Exosomes 50 nm	-	No changes in exosome morphology or size Change in exosome surface charge ↓ Viability in HepG2 and U87 co-cultivated with exosomes Cellular uptake of exosomes HepG2 and U87 ↑ ROS in HepG2 co-cultivated with exosomes

^1^ A549 (human lung epithelial adenocarcinoma cancer cells), BMSCs (Bone mesenchymal stem cells), Cas (Caspase), CAT (Catalase), HepG2 (Human hepatocellular carcinoma), HMVECs (human microvascular endothelial cells), HUVECs (Human umbilical vein endothelial cells), iNPCs (Neural progenitor cell), LDH (Lactate dehydrogenase), L02 (Human embryonic liver cell), RAW264.7 (Macrophage-like), MDDC (Monocyte-derived dendritic cells), OVA (Ovalbumin-sensitized), mESCs (mouse embryonic stem cells), PBMC (Primary human peripheral blood mononuclear cells), PMQ (non-activated primary mouse peritoneal macrophages), Tc (T cytotoxic cell), Th (T-helper cell), Th1 (T-helper cell type 1), Th2 (T-helper cell type), THP-1 (Human leukemia monocyte-like), VSMCs (primary mice vascular smooth muscle cells). ^2^ AuNPs (Gold nanoparticles), AgNO_3_ NCs (Silver NCs), CaP (calcium phosphate particles), COOH-terminated (anionic), Fe_3_O_4_ NCs (Iron oxide NCs), Fe_3_O_4_NPs (Magnetic iron oxide nanoparticles), MIONs (Magnetic iron oxide nanoparticles), NCs (Silver and Iron oxide nanoclusters), nHAp (Nano-hydroxyapatite), NH2-terminated (cationic), NPs (Nanoparticles), PAMAM (polyamidoamine dendrimers), PEI-SPION NPs (NPs superparamagnetic iron oxide NPs associated with NPs polyethyleneimine), Pd NPs (Palladium nanoparticles). POSS NPs (Polyhedral oligomeric silsesquioxane nanoparticles), Pt NPs (Platinum nanoparticles), s-GO (Small graphene-oxide nano-flakes), SiO_2_NPs (Silicon dioxide nanoparticles), SWNCTs (acid-oxidized single-walled carbon nanotubes), TiO_2_ NPs (Commercial titanium dioxide), ZnO (Commercial zinc oxide). ^3^ UC (Ultracentrifugation), TEM (Transmission electron microscopy), NTA (Nanoparticle Tracking Analysis), SEM (Scanning Electron Microscopy), FC (Flow cytometry), HSC (High-speed centrifugation), MS (Magnetic separation), CLSM (Confocal Laser Scanning Microscopy), WB (Western blot), AFM (Atomic force microscopy), UVRR (UV Resonant Raman), FTIR-ATR (Attenuated Total Reflection Fourier Transform Infrared), AchE (Acetylcholinesterase), EDS (Energy-Dispersive X-ray Spectroscopy), SP-ICP-MS (Single Particle Inductively Coupled Plasma Mass Spectrometry), ICP-OES (Inductively Coupled Plasma Optical Emission Spectrometry), FACS (Fluorescence-activated cell scanning), QCM-D (Quartz crystal microbalance with dissipation), LC-MS/MS (Liquid chromatography-tandem mass spectrometry), FP (Fluorescence polarization), BCA (Bicinchoninic acid assay kit), PEG (Polyethylene glycol), RT-PCR (Reverse transcription polymerase chain reaction), qRT-PCR (Quantitative Reverse Transcription-Polymerase Chain Reaction); ^4^ MVBs (Multivesicular bodies), EVs (Extracellular vesicles), MVs (Microvesicles), EVs-5 (Extracellular vesicles); ^5^ CD (Tetraspanin), TSG101 (Tumor susceptibility gene 101); ^6^ CD (Tetraspanin), TSG101 (Tumor susceptibility gene 101), IL (Interleukin), PS+ (phosphatidyl serine-positive), PSCs (Heterogeneous postsynaptic currents), U87 (Human primary glioblastoma), Ca (Calcium), BALF (Bronchoalveolar lavage fluid), iDCs (Immature dendritic cells), APCs (Antigen-presenting cells), MHC (Major histocompatibility complex), AM φ (Alveolar macrophages), DC1 (cytokine DC subset 1), Erk (Extracellular regulated protein kinase), HSP70 (Heat shock protein 70).

## 4. Discussion

Exposure to toxic and sub-toxic concentrations of NMs can trigger several (patho)physiological effects, leading to dysfunction in human health. Therefore, analyzing the interaction and behavior of NMs in biological environments is an urgent need. This study focused on research that demonstrated a potential relationship of EV secretion induced by NMs. EVs are now recognized as an important pathway for intercellular communication by exchanging various types of payloads between cells [21]. While the cellular transfer of NMs and their payloads has received increasing attention, the role of EVs in this process remains elusive. It is likely that upon cell internalization, some NMs will unwittingly end up in multivesicular bodies (MVBs) as well as within EVs, inducing specific biological consequences (see Figure 2).

### 4.1. Effects of NMs on EVs Biogenesis

As already reported in the literature and observed in this ScR, NMs of different chemical compositions interact with cells, organs, and tissues, causing several changes that include alterations in cell viability, activation of inflammatory processes, and disturbance of EV secretion. A map of the main interconnections among the main results of the 18 articles is represented in Figure 3A, and maps of the main biological outcomes are shown in Figure 3B. As already known, EVs are generated through complex biological mechanisms and are responsive to several physical and chemical factors. Several studies demonstrated an increase in exosome secretion, activity, composition, surface markers, and intravesicular proteins upon intrinsic and extrinsic stimulation, including pH variations, oxidative stress, hypoxia, cholesterol, cell detachment, cell type, media, and concentration of serum in the media, or Ca2+ ionophores [52]. While cellular stress is well known to significantly impact the vesicular secretome, NM-induced oxidative stress-elicited changes or release of exosomes have not been completely elucidated.

In this ScR, we found that the variety of NMs exposed to the different biological models increased exosome secretion (see Figure 3B), whereas no significant differences were observed regarding their size or morphology [26,39,40,41,44,45,46,49,51]. Many studies reported vesicles with a cup-shaped appearance in the negative-stain TEM with HSP 70, CD63, CD81, CD9, LAMP1, and TSG101 as characteristic markers, suggesting that the vesicular secretome that was isolated also comprised exosomes. The results also demonstrated that exosome secretion upon NM stimulation was independent of the cytotoxicity of the NM. Just like the different physicochemical characteristics, no NM can be deduced as a determinant for the stimulation of EV secretion, because materials of different chemical compositions, size, crystal structure, morphology, and surface charge ended up stimulating secretion upon NM exposure. Although cell viability upon NM exposure (SWNCTs, PEI-SPIONs, s-GO NMs, POSS, nHAp, Au, CaP, Fe_3_O_4_, and Pt nanoparticles) was high, the number of exosomes increased compared to the control group. One example is human lung A549 cells that were treated with platinum nanoparticles. Here, the induction of exosome secretion was demonstrated through oxidative stress and ceramide pathways [44].

Nonetheless, further studies are necessary to elucidate other potential mechanisms for the observed elevation in EV secretion. Interestingly, even when NMs (e.g., palladium nanoparticles and polyamidoamine dendrimers) were cytotoxic, EVs were released in high numbers [45,50]. Although the induction of EV release by highly toxic NMs requires further study, exosomes release was reported to be strongly correlated with oxidative stress, apoptosis, and immunomodulation in the case of palladium nanoparticles [45]. It has been suggested that NM-mediated oxidative stress induces autophagy, enhancing the formation and accumulation of MVBs within the cells, resulting in an increased release of exosomes before the cells enter apoptosis [26].

Subtoxic concentrations of TiO_2_ or ZnO nanoparticles exposed to macrophages and dendritic cells did not alter exosome secretion [39], as well as Au nanoparticles (NPs) that did not change macrophage-derived exosome biogenesis. At first, it was reported that due to the small size of Au NPs, they were able to be internalized by exosomes, and this proved to be a route of Au nanoparticle elimination (see Figure 3B) [43]. It was found that the loading of EVs with NMs is typically dependent on the surface characteristics of the NMs and the amount of material delivered to the cell, which is, in turn, governed by the strength of the material-cell interaction. For instance, NMs with little or no cell-binding capability will rely upon weak, nonspecific interactions with the cytoplasmic membrane. In this situation, a higher NMs concentration and prolonged incubation time are required to maximize EV loading. However, it also depends on the phagocytosis and uptake mechanisms that differ between phagocytes and non-phagocytic cells. The NM surface parameters (e.g., size and surface coating) determine cellular uptake and subsequent sorting into EVs. Another exciting aspect is that EVs can also mediate the intercellular exchange of nanoparticles [53]. Theoretically, transcellular transport consists of four steps: entry into one cell, intracellular transport, cargo export or exocytosis, and the re-entry of released cargo into another cell. Preliminary results showed that cell-penetrating peptides (CPP)-loaded AgNPs can be exocytosed freely or enclosed inside EVs. In vitro and in vivo studies revealed that EVs account for a significant fraction of intercellular exchange and transport. Intriguingly, while freely released nanoparticles engage with the same cellular receptors for re-entry, EV-enclosed ones bypass this dependence. These studies provided an easy and precise system to investigate the intercellular exchange stage of NMs delivery and shed the first light on the importance of EV transport between cells and across complex tissues [53]. Trojan exosome hypothesis where virus uses exosome biogenesis pathway for the formation of infectious particles. Exosomes hijack viruses to use as trojan horses for the cell-to-cell spreading of infectious particles while escaping the immune system [54].

As MVBs are the cellular factory responsible for exosomes biogenesis, some authors focused on the process of MVB formation within the EV-producing cell rather than the EV secretome, in order to analyze changes in EV biogenesis upon NM stimulation. During exosome biogenesis, invagination of the late endosomal membrane gives rise to the formation of intraluminal vesicles (ILVs). In a complex mechanism controlled by the ESCRT (endosomal sorting complexes required for transport) protein machineries, specific proteins and cytosolic components (lipids, nucleic material including DNA, mRNA, microRNA, small-interfering RNA) are sorted for inclusion in ILVs [52]. At this stage, NMs may coincidentally end up within the newly formed ILVs; however, the mechanisms remain elusive. Finally, the fusion of MVBs with the plasma membrane releases their content of ILVs as exosomes into the extracellular space [55]. In this ScR, eight articles indeed demonstrated that NMs were internalized in MVBs [20,26,39,43,44,47,48]. It is conceivable that the internalization of NMs in MVBs possibly interferes with the process of exosome biogenesis. However, it is unknown how NMs may activate or modulate the exosomal biogenesis pathways. It is important to understand that the release of these EVs does not occur by chance but through selective interactions, stringently controlled by ESCRT and other control factors, thus ensuring a highly controlled exchange of molecular information between cells [55].

In most studies, NMs did not alter the size of EVs except for Au NPs. For the same size of particles, one study reported an increase in the average size of primary human peripheral blood mononuclear-derived EVs due to Au-NP internalization [25]. In contrast, significant differences in the diameter of mouse embryonic stem-cell-derived EVs were observed [43]. This divergence may be related to the cell model, the size of the vesicles or the agglomeration state of NMs in the biological media. However, one cannot exclude the possibility that different characterization techniques were employed. Following the MISEV guidelines, different but complementary techniques are not interchangeable in their implementation since all techniques have limitations. For example, EVs > 400 nm or <50 nm are not well detected by all types of NTA instrumentation. For this reason, EV-size-distribution analyses are better interpreted when combined with cryo or conventional electron microscopy (e.g., TEM) or super-resolution microscopy techniques. These are gold standards for unbiased size determinations at the single-vesicle level; however, they need to be performed on sufficient sample sizes in order to deliver statistically representative data on heterogeneous EV populations.

### 4.2. Effect of NMs on EV Cargo

In this ScR, we observed that few studies explored the molecular cargo of EVs after exposure to NMs [44,45]. The studies included here, [24], revealed that vesicular miRNAs involved in increased phagocytosis of inhaled SiO_2_ NPs were differentially expressed in patients with pneumoconiosis compared to healthy individuals. The SiO_2_NPs generated a suppression of hsa-let-7a-5p and has-let-7i-5p miRNA in macrophage- and fibroblast-derived EVs, thereby increasing the activity of the WASL gene and generating the formation of the WASL and VASP complexes, which consequently increased the Arp2/3-induced phagocytosis of SiO_2_NPs. These processes may be a new pathophysiological mechanism of pneumoconiosis and indicate that EVs (modified by NMs) may be the key to disease progression and severity [24]. Consequently, these extracellular miRNAs may represent new biomarkers of pneumoconiosis, opening the potential for improving diagnosis and providing a better prognosis to patients [56].

Moreover, Fe_3_O_4_ NPs altered miRNA release via EVs, as increased miR-21-5p levels could be observed in bone mesenchymal stem cells [40]. The miR-21-5p is an important miRNA involved in cell migration and wound healing, as well as resulting in increased cell proliferation [57]. It has been described as a cell-growth activator in colon cancer [58], esophageal cancer [59], and ovarian cancer [59], promoting blood-vessel formation [60]. In addition, miR-21-5p increases resistance to cisplatin treatment of lung cancer [57]. Cell migration and increased proliferation were confirmed by performing functional assays with EVs enriched with miR-21-5p on HUVECs, human and mouse skin fibroblasts, demonstrating that miR-21-5p-exosomal positively regulated pro-angiogenic and pro-fibrogenic genes such as VEGF, HIF-1α, PDGFRα, and bFGF [40]. This upregulation of miR-21-5p-release may be an essential mechanism to promote skin regenerative effects, and its silencing may represent a new treatment and better cancer prognosis [58,59,61]. Functional studies demonstrated that miR-21-5p silencing induces a G0/G1 arrest of the cell cycle and, consequently, decreases cell proliferation in colon adenocarcinoma [58].

Regarding EV protein content after stimulation by NMs, [44], it was found that AuNPs enrich EVs (derived from mouse embryonic stem cells) in specific proteins, which the authors proposed as a potential new approach for cancer treatment. Among the 13 enriched proteins found in EVs, they also detected ribosomal proteins (60S ribosomal protein L27a, 60S ribosomal protein L3, 60S ribosomal protein L6), previously described as inhibitors of cancer-cell proliferation [62,63,64]. The particles did not induce the expression of cofilin or p-Erk (involved in breast-cancer-metastasis processes), which supported their use as a potential future cancer treatment. Additional proteins found were involved in central signaling pathways related to the regulation of cell transport, DNA/RNA binding and regulation of catalysis, metabolic processes, and inflammation. An increase in flotillin-1 in EVs derived from primary neural cultures after treatment with graphene-oxide nanosheets was observed [50]. Synthetic hydroxyapatite nanoparticles were found to increase calcium levels of vascular smooth-muscle-derived EVs [48], with this process possibly leading to vascular calcification [65,66,67,68]. Basically, phosphate and calcium are transformed to nano-hydroxyapatite by alkaline phosphatase inside EVs. This process is even intensified by exposure to hydroxyapatite, as studies have shown that an excess of these crystals stimulated extracellular-matrix calcification and activated the osteogenic pathway of cells in the vascular wall [49,65]. Hence, EV proteins and miRNAs have been identified as potential cancer biomarkers, as well as implicated in osteoarthritis, rheumatoid arthritis, inflammatory bowel diseases, neurodegenerative pathologies, complex regional pain syndrome, and peripheral nerve damage [69,70]. Possibly soon, miRNAs and proteins derived from EVs may act as a bridge between exposure to NMs, their pathophysiology, and diagnosis [71].

### 4.3. Pathophysiological Implications

As we have learned from the original studies discussed in the results section of this ScR, NM exposure generally enhances the production and secretion of EVs and modulates their content. In the context of human health, this may result in (i) efficient elimination of NMs, but also alterations (ii) of cellular functions and biological activity, (iii) in cell communication, (iv) as well as immunomodulation, inflammation, and immune activation, or (v) angiogenesis and healing through the activation of specific pathways may be observed as overviewed in Figure 4. By applying objective methodologies, this ScR provided evidence that, firstly, macrophages may play an important role in eliminating NMs through sorting into EVs, as they scavenge the body of both organic and inorganic substances by the release of high numbers of EVs. NMs can be exported inside EVs, and EVs can be transported into another cell with their enclosed particulate matter. Hence, they can serve as the membrane-enclosed conduit for particle elimination and intercellular exchange.

The involvement of EVs induced by NM exposure with impact on cellular functions and resulting pathophysiological outcomes was highlighted by several studies that were objectively selected for this ScR. For instance, we found published evidence that microvesicles derived from mouse embryonic stem cells after NM exposure attenuated 4T1 tumor-cell proliferation and migration through inhibition of cofilin expression and extracellular-regulated-protein-kinase (Erk) phosphorylation [43]. These alterations may provide a strategy for reduced tumor-cell metastasis since the tumor uses EVs to attract stem cells and re-program their functional profile to be pro-tumorigenic [72]. Likewise, NM-exposure-induced, astrocyte-derived microvesicles were shown to modulate basal synaptic transmission, inducing a stable increase in synaptic activity accompanied by changes in neuronal plasma-membrane elastic features [50]. These outcomes are regulated by increased heterogeneous postsynaptic currents and decreased neuronal rigidity, thereby facilitating synaptic vesicle release and regulating neurotransmission. Thus, the regulation of cellular functions using EVs that were induced by graphene-oxide nano-flake exposure seems to be an approach that holds the potential for new opportunities for pharmacological intervention in neurodegeneration [73]. Macrophage-derived EVs induced by NM exposure increased the transdifferentiation of fibroblasts into myofibroblasts, in addition to increasing collagen deposition [24], representing a key molecular step in developing organ fibrosis [74].

The exchange of microparticles between endothelial cells is increased by cell stress [75]. NMs induced the increased secretion of endothelial-cell-derived EVs loaded with nanoparticles that were consequently taken up by HMVECs, 4T1 or MCF-7 [20]. The exchange of endothelial-cell-derived EVs may promote the transmission of pro-inflammatory, pro-coagulant, and pro-apoptotic biological messages [76,77,78].

This ScR further compiled evidence that NMs promote immunomodulation mediation via the action of EVs, which is a process that involves the activation, differentiation, proliferation, and maturation of immune cells such as macrophages, dendritic cells, and lymphocytes [22,41,45]. EVs derived from cells exposed to NMs play a crucial role as modulators of classic macrophage activation (pro-inflammatory subset M1) [79], maturation of DCs characterized by upregulation of surface markers and cytokine secretion, as well as T-cell activation and polarization, which is a requirement for detecting a real deviation in the immune state from a momentary fluctuation in tissue homeostasis [79]. Nevertheless, these studies suggest that the stimulation of systemic adaptive immunity can be mediated via NM-induced vesicles. This suggests that EVs are effective mediators of spatially distant immune activation and inflammatory responses that can often be observed upon exposure to NMs. 

Finally, EVs can orchestrate healing and tissue regeneration, mainly through the transfer of signaling molecules also as potential drug delivery systems (DDS) [18,80,81,82]. Although there are already several studies indicating the potential use of NMs to stimulate regeneration, little is known about the effects of NM exposure-induced EVs in this process. The results of the studies included in this ScR indicated that EVs secreted upon exposure to NMs contribute to healing and skin regeneration [26,40]. In addition to promoting viability, migration, and increased expression of pro-angiogenic (VEGF, HIF-1α, HSP-70, Ang-1, and Ang-2) and pro-fibrogenic genes (PDGFRα and bFGF), the exposure to NMs induced the upregulated expression of vesicular miRNA-21 and miRNA-155 with the inhibition of SPRY2 and the activation of the PI3K/AKT and ERK1/2 signaling pathways [78,79,80]. The upregulation of secreted miR-21-5p might potentially improve wound healing in the skin, mainly for patients with diabetes who are suffering from extensive wounds.

We hypothesize that the impact of NMs on cellular communication performed by EVs may be the key to the potential future development of therapeutic and diagnostic strategies based on using EVs as biomarkers in diverse diseases following toxic NM exposure. However, it will be crucial to undertake all necessary efforts to standardize the available methodologies for isolation and characterization of EVs supplemented by comprehensive documentation and implementation of validated protocols.

### 4.4. Challenges and Limitations on EV Biogenesis Studies upon NMs Exposure

Published studies on the safety assessment of NMs and their possible cytotoxic impacts vary substantially, sometimes promoting controversy in the scientific discourse and concomitantly causing regulatory uncertainty. As we encountered during the careful literature review in preparation of this ScR, this is possible because many studies do not adequately describe the methodology (in vitro, in vivo, ex vivo, and clinical) applied. Specifically, there was generally an absence or significant paucity of physicochemical characterization of NMs in the respective biological environment combined with a lack of metadata reporting [80]; a variety of dose metrics was used, and no standardized methods were applied [51,80]. In particular, the doses of NMs administered and the time points at which the EV secretion was assessed were very heterogeneous, and many studies did not provide a comprehensive characterization of EVs (including protein markers, morphology, and concentration/count measurements) [83,84]. To improve the comparability between studies, the selection of cells, incubation time, serum concentrations, pH, temperature, NM dose, and other parameters must be uniformly monitored to avoid any variations. Another limitation observed was the methodological quality reported in the toxicological studies. Only four studies were considered reliable without restrictions, classified into both categories (A and B) in the ToxRTool [39,44,50]. This demonstrates that most of the studies were not concerned about clearly and authentically reporting toxicological results, which are highly relevant to studies involving highly heterogeneous species such as EVs where the isolation process determines the product. Although the MISEV guideline provides excellent recommendations for reporting EV-isolation methodology, EV nomenclature, minimal analytical characterization, and functional studies, only two studies performed the registration on the EV-TRACK platform, with an EV-METRIC being assigned [22,39]. It is important to note that this ScR did not exclude any study based on a quality assessment in order to enable a broader overview of the possible effects exerted by EVs upon exposure to NMs. Finally, it has to be noted, as the search strategy was limited to the English language, there may exist studies in other languages not included in this ScR. However, given the high number of studies included in the first stage of selection (2944 unique results), and the identified heterogeneity in the quality of studies in this field, it is likely that additional non-English language studies would have altered the results significantly.

## 5. Conclusions

This ScR overviews the current evidence that NMs can alter EV biogenesis and cargo, thus affecting cell communication and tissue homeostasis. The stimulation of EV biogenesis by NMs seems to be a phenomenon independent on the physicochemical characteristics of NMs; however, it has emerged observation that it this outcome may depend on the biological model used. Results demonstrated that specific NMs altered the EV molecular cargo, which even endowed them with new activities to stimulate angiogenesis, healing, inflammation, and immune activation. Small-sized and biocompatible Au-NP cells seemed to use EVs as a route of NM elimination. At present, the mechanisms underlying EV biogenesis upon NM exposure and sorting of these co-transported molecules remain to be investigated in greater detail. One of the major gaps to be addressed is how the NM properties and doses affect EVs secretion, cargo, trafficking, and cellular uptake. Remaining questions include: What are the energetic requirements for EV biogenesis? What are the molecular mechanisms controlling the sorting of NMs into MVBs and, consequently, exosomes and microvesicles? Can NMs be engulfed into MVBs to alter EV cargo, and, if so, how? Thus, understanding the roles of EVs as mediators of intercellular communication and cargo delivery will help to reveal other adverse-outcome pathways in the toxicology of NMs, since EVs can cross the endothelial barrier and represent a means to disseminate their cargo to and broadcast signals to distant locations. Based on these findings, EVs may emerge as a fundamental tool to reveal the various mechanisms by which NMs exert their impact and pave the way to possibly identify other cell and tissue interactions of NMs that as yet remain to be explored.

## Figures and Tables

**Figure 1 nanomaterials-12-01231-f001:**
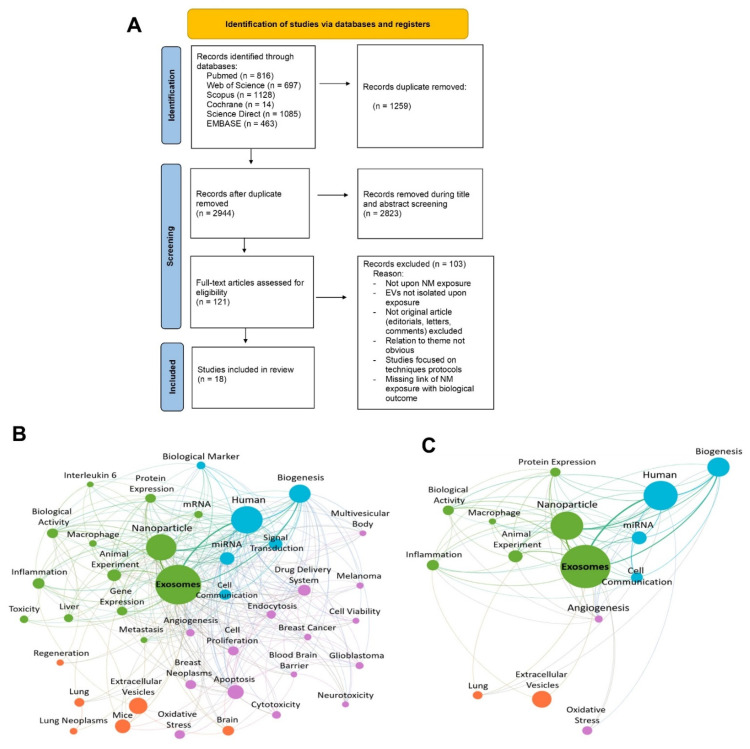
(**A**) PRISMA flow diagram applied in this ScR, co-occurrence network of the most frequently used keywords in the (**B**) 2944 unique results and (**C**) 18 ultimately selected articles. Each color suggests a set of keywords shared among the analyzed articles.

**Figure 2 nanomaterials-12-01231-f002:**
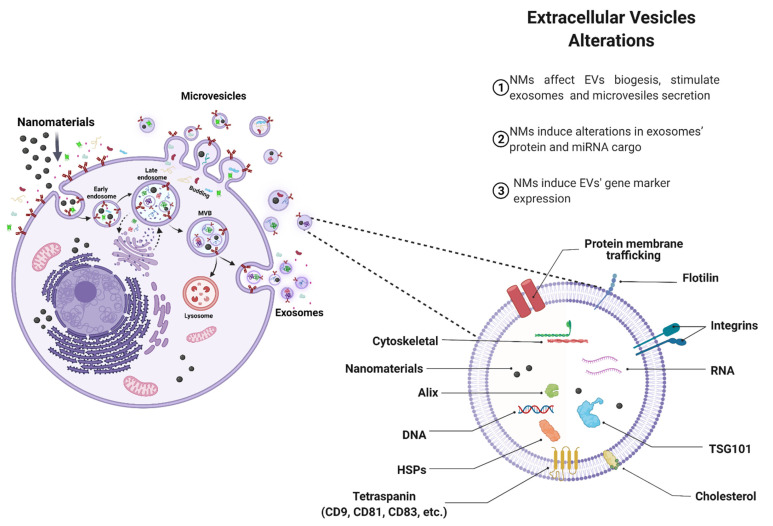
Schematic representation of the overall effect of NMs on secreted EVs.

**Figure 3 nanomaterials-12-01231-f003:**
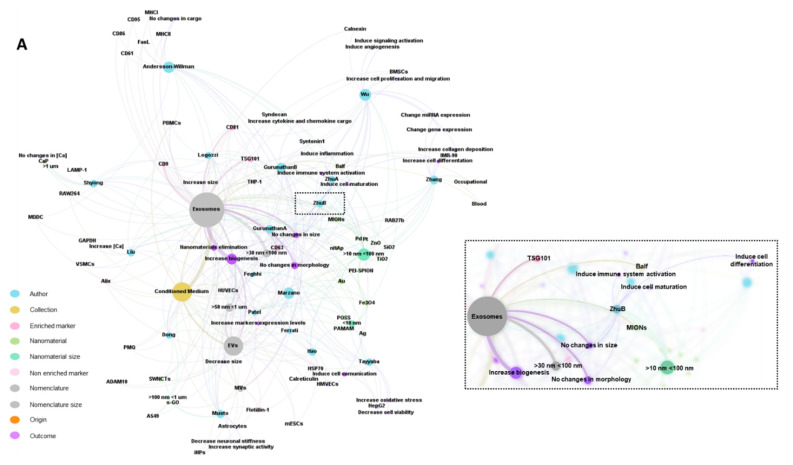
(**A**) Detailed network of the 18 articles that met all the eligibility criteria. The dashed rectangle highlights article [41], summarized in Table 3. In short, [41] used MIONs with a size ranging between 10 and 100 nm. MIONs were exposed to a bronchoalveolar lavage fluid (BALF) extracted from mice, inducing an increase of EVs biogenesis. EVs, with a size between 30 and 100 nm and enriched in TSG101, induced iDC maturation, immune-system activation, and T-cell differentiation. Overall, and regarding all experimental conditions, exosome morphology and size were not affected. (**B**) Most relevant biological outcomes and their connections among the 18 analyzed articles. For instance, despite using different experimental conditions, 15 out of 18 articles pointed to increased EV biogenesis as one of their outcomes. Regarding the duplicated author’s nomenclature, ZhuA, ZhuB, GurunathanA, and GurunathanB correspond to [22,41,44,45], respectively.

**Figure 4 nanomaterials-12-01231-f004:**
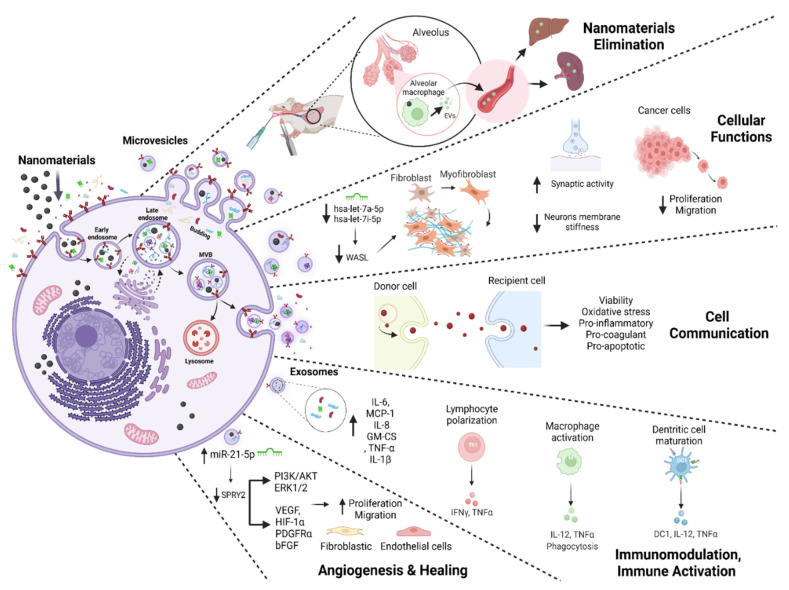
Cell exposure to NMs induces EV biogenesis and modulations in various biological activities: elimination of NMs; impact on cellular functions; alterations in cell communication; immunomodulation, inflammation, and immune activation; and angiogenesis and healing.

**Table 1 nanomaterials-12-01231-t001:** Definition of key terms based on the PECO strategy [30].

Acronym	Definition	Description
P	Population	Extracellular vesicles
E	Exposure	Nanomaterials
C	Comparison	Extracellular vesicles without nanomaterials exposure
O	Outcome	Cellular response

## Data Availability

All datasets generated for this study are included in the article/Appendix A.

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
