# Peer review of "Nanomaterial Exposure, Extracellular Vesicle Biogenesis and Adverse Cellular Outcomes: A Scoping Review"

_nanomaterials, 2022, doi:10.3390/nano12071231_

Round 1

Reviewer 1 Report

It was complex content to understand completely due to the huge data of analysis. This review is probably useful for the relation of nanomaterials and cell response including EV biogenesis. I enjoyed the reading. 

Fig 1 was hard to see due to the small characters.

I recommend that Table 2&3 should be concentrated because it consumes 5 pages. This is not good for future reader (I agree the information in the Tables).

Author Response

The authors would like to express their gratitude to the editor and reviewers for considering our manuscript and providing valuable suggestions to improve its quality. A detailed reply to the comments and concerns of the reviewers is presented below. All page numbers refer to the revised manuscript with modifications marked up using the “Track Changes” function.

Point 1: Fig 1 was hard to see due to the small characters.

Response 1: We agree with the reviewer comment, thus we changed the font and figure size.

Point 2: I recommend that Table 2&3 should be concentrated because it consumes 5 pages. This is not good for the future reader (I agree with the information in the Tables).

Response 2: We understand that the size of the tables can become tiresome for readers. Efforts were done to synthesise information, however, it was not possible to concentrate Tables 2 and 3, since from our point of view they contain essential information for the reader to understand the manuscript. Therefore, we reduced the fonts size of Tables 2 and 3 for 8 pt., the smallest size allowed by the journal.

Reviewer 2 Report

The review by Lima et al is an excellently structured and written manuscript which summarizes an interesting and important field in which way nanomaterials influence the extracellular vesicle biogenesis. This topic connects the different research areas of EVs which either looking into biomarkers for diseases or using them as drug delivery vehicles and to lesser extend how nanomaterials as pollutant or drug delivery system influence the EV generation and composition. To get this particluar information is always very time consuming. This review nicely summarizes certain effects of various nanomaterials on the EVs and subsequent cellular effects. Based on the fact, that the authors started with ~3000 papers and due to strong eligibility criteria ended in only 18 manuscripts met all criteria clearly shows that this topic is just at the beginning.

Generally, there is nothing negative or improvements I can mention. The authors pointed out in detail how they started their search and with which quality criteria they came to the final result. However, one has to ask if some exclusion criteria were set to stringent and some interesting results got lost?

The results are clearly summarized in tables and discussed in the following text. I really like the authors’ style as they commented key findings from their literature search and the included papers and thus sharing their knowledge in a nice communicative and scientific way. Also, very nice is, how they conclude and connect the results from the papers for possible therapeutic interventions or toxicological characterizations of nanomaterials for the future. This review is written not only for EV scientists but also for other researches working in the field of nanoparticle toxicity and drug delivery.

Just one comment, in line 357 is a typing error “DC9” it should be “CD9”.

This review would like to see published!

Author Response

The authors would like to express their gratitude to the editor and reviewers for considering our manuscript and providing valuable suggestions to improve its quality. A detailed reply to the comments and concerns of the reviewers is presented below. All page numbers refer to the revised manuscript with modifications marked up using the “Track Changes” function.

Point 1: The authors pointed out in detail how they started their search and with which quality criteria they came to the final result. However, one has to ask if some exclusion criteria were set to stringent and some interesting results got lost? 

Response 1: The complete list of eligibility criteria (exclusion and inclusion criteria) can be seen by reviewers and readers in Supplemental Methods (see Table S2, which can be downloaded at: https://1drv.ms/w/s!Ak26ijFbMAx8xSNAFhEORSSVwSa3?e=gu8DGb. We agree with the reviewer that some interesting results can be lost during the selection process, however, all efforts were done to overcome this problem. The use of pre-specified and explicit exclusion criteria stringent minimizes the risk of selection bias, besides ensuring that the inclusion or exclusion of articles is done in a transparent manner, and as objectively as possible [1–3]. Moreover, not using consistent eligibility criteria or using less stringent criteria can lead to inconsistent conclusions [4]. Another important consideration is that our eligibility criteria reflect the question being asked and therefore follow from the ‘key elements’ that describe the question structure [4]. We need to stress that our evidence selection decision process was conservative at each step, being performed by four reviewers blind (LG, PS, TL, and WS) and disagreements between the reviewing authors were solved through careful discussion, and any remaining disagreements were solved by a fifth reviewer (ARR), decreasing the risk of eligible studies being lost. This is considered a very complete and eficient method, since a minimum of two screeners is now considered mandatory in some systematic and scoping reviews [1,3].

[1]        Peters MDJ, Godfrey CM, Khalil H, McInerney P, Parker D, Soares CB. Guidance for conducting systematic scoping reviews. International Journal of Evidence-Based Healthcare 2015;13:141–6. https://doi.org/10.1097/XEB.0000000000000050.

[2]        Levac D, Colquhoun H, O’Brien KK. Scoping studies: advancing the methodology. Implementation Science 2010;5:69. https://doi.org/10.1186/1748-5908-5-69.

[3]        Cumpston M, Li T, Page MJ, Chandler J, Welch VA, Higgins JP, et al. Updated guidance for trusted systematic reviews: a new edition of the Cochrane Handbook for Systematic Reviews of Interventions. The Cochrane Database of Systematic Reviews 2019;10:ED000142. https://doi.org/10.1002/14651858.ED000142.

[4]        Colquhoun HL, Levac D, O’Brien KK, Straus S, Tricco AC, Perrier L, et al. Scoping reviews: time for clarity in definition, methods, and reporting. Journal of Clinical Epidemiology 2014;67:1291–4. https://doi.org/10.1016/j.jclinepi.2014.03.013.

Point 2: Just one comment, in line 357 is a typing error “DC9” it should be “CD9”.

Response 2: We thank this reviewer for highlighting this issue, so we have fixed this error in line 379 in the revised version.

Reviewer 3 Report

This is a good overview, from the choice of topic to the depth of discussion. This manuscript deserves to be published in Nanomaterials. The following suggestions may help the authors to further improve the quality of this manuscript.

  1. What criteria are the authors using to determine papers that are of sufficient quality to include in the review?
  2. The expression of the author about the physicochemical characteristics of the nanomaterials in the text is too general and should be specific about what properties play a major role.
  3. The authors focus on concentration effects, but rarely describe the range of concentrations studied specifically in the text. Further clarification of the issue is needed.
  4. Are experimental exposure conditions an influential factor? The authors need to discuss this further with the context of the literature.
  5. Do the physicochemical properties and concentrations of nanomaterials change before and after they enter the cell? The authors need to take note of this question and elaborate further.

Author Response

The authors would like to express their gratitude to the editor and reviewers for considering our manuscript and providing valuable suggestions to improve its quality. A detailed reply to the comments and concerns of the reviewers is presented below. All page numbers refer to the revised manuscript with modifications marked up using the “Track Changes” function.

Point 1: What criteria are the authors using to determine papers that are of sufficient quality to include in the review?

Response 1: Authors are aware of the need for quality criteria to determine the reliability of the evidence produced. However, it is important to stress that even though systematic review share several common characteristics (namely, collecting, evaluating, and presenting the available research evidence with systematic, transparent, and replicable) with scoping review (ScR), the methodology of scoping review typically doesn't include a process of quality assessment. The objective of a ScR is to map relevant literature and try to address the current gap in knowledge in the field of interest [1–3]. In that way, we don't use articles quality as determining factor to include or exclude papers in the review. Nevertheless, we understand the benefits of having quality criteria and that's why we include in this ScR the following tools: ToxRTool [4], ARRIVE Guideline 2.0 [5], Newcastle-Ottawa Scale (NOS) [6] and MISEV (EV-TRACK) [7]. The goal was to analyse the quality of the studies and data included in this scoping review. The information and assessment were clarified in detail in Section 4.4 Challenges and limitations on EV biogenesis studies upon NMs exposure (line 674 of the revised manuscript).

Point 2: The expression of the author about the physicochemical characteristics of the nanomaterials in the text is too general and should be specific about what properties play a major role.

Response 2: The physicochemical characteristics of NMs have an biological impact on toxicity and adverse biological outcomes and several articles has been published during the past decades. However, no clear trends for a specific type of NM, surface functionalization, or physicochemical characteristics (size, crystalline structure, surface charge, etc.), can be deduced as the determinant for EVs stimulation. For example, different formulations of iron oxide nanoparticles with different physicochemical characteristics (crystalline structure, morphology, size and surface charge) have induced EVs and exosomes secretion despite having different adverse biological outcomes as can be seen in table 1 [8–11]. We introduce in the manuscript two sentences clarifying this pouint (in line 449 from the revised file).

Point 3: The authors focus on concentration effects, but rarely describe the range of concentrations studied specifically in the text. Further clarification of the issue is needed.

Response 3: The range of concentrations reported in the 18 articles is described in Table 2 on "Exposure conditions". As it can be see the range of concentrations is diverse limiting the comparison among studies.

Point 4: Are experimental exposure conditions an influential factor? The authors need to discuss this further with the context of the literature.

Response 4: We thank this reviewer for highlighting this issue! In specific in nanosciences, the experimental conditions have an impact and there in fact is plenty of resources to be mentioned for that. We thus further extended on protein corona formation as the most genuine molecular initiating event regarding this issue, and to realistic in vitro dosing related to in vivo observed conditions as another one citing an overviewing reference that published recommendations on how to perform such types of assays. However, the studies we refer to here are less making a point about it, so we have now included a critical remark on that in the revised version. Some alterations were performed in the article to explain that the NM experimental exposure condition biological does not impact the observed results also that no clear trends for a specific type of NM. We have included a statement to clarify this in the revised ScR version (see please 291).

Point 5: Do the physicochemical properties and concentrations of nanomaterials change before and after they enter the cell? The authors need to take note of this question and elaborate further.

Response 5: We are also very grateful to the reviewer for bringing this up which we tried to tackle in harmony with the previous comment citing a recently published overview article giving recommendations for in vitro experimentation in nanosafety assessment [19]. The studies reviewed in the ScR did not investigate the concentration effects of NM upon uptake.

[1]       Arksey H, O’Malley L. Scoping studies: towards a methodological framework. International Journal of Social Research Methodology 2005;8:19–32. https://doi.org/10.1080/1364557032000119616.

[2]        Grant MJ, Booth A. A typology of reviews: an analysis of 14 review types and associated methodologies. Health Information & Libraries Journal 2009;26:91–108. https://doi.org/10.1111/j.1471-1842.2009.00848.x.

[3]        Levac D, Colquhoun H, O’Brien KK. Scoping studies: advancing the methodology. Implementation Science 2010;5:69. https://doi.org/10.1186/1748-5908-5-69.

[4]        Schneider K, Schwarz M, Burkholder I, Kopp-Schneider A, Edler L, Kinsner-Ovaskainen A, et al. “ToxRTool”, a new tool to assess the reliability of toxicological data. Toxicology Letters 2009;189:138–44. https://doi.org/10.1016/j.toxlet.2009.05.013.

[5]        Percie du Sert N, Hurst V, Ahluwalia A, Alam S, Avey MT, Baker M, et al. The ARRIVE guidelines 2.0: Updated guidelines for reporting animal research. British Journal of Pharmacology 2020;177:3617–24. https://doi.org/10.1111/bph.15193.

[6]        Hartling L, Milne A, Hamm MP, Vandermeer B, Ansari M, Tsertsvadze A, et al. Testing the Newcastle Ottawa Scale showed low reliability between individual reviewers. Journal of Clinical Epidemiology 2013;66:982–93. https://doi.org/10.1016/j.jclinepi.2013.03.003.

[7]        Théry C, Witwer KW, Aikawa E, Alcaraz MJ, Anderson JD, Andriantsitohaina R, et al. Minimal information for studies of extracellular vesicles 2018 (MISEV2018): a position statement of the International Society for Extracellular Vesicles and update of the MISEV2014 guidelines. Journal of Extracellular Vesicles 2018;7:1535750. https://doi.org/10.1080/20013078.2018.1535750.

[8]        Wu D, Kang L, Tian J, Wu Y, Liu J, Li Z, et al.

Exosomes Derived from Bone Mesenchymal Stem Cells with the Stimulation of Fe3O4 Nanoparticles and Static Magnetic Field Enhance Wound Healing Through Upregulated miR-21-5p

. International Journal of Nanomedicine 2020;Volume 15:7979–93. https://doi.org/10.2147/IJN.S275650.

[9]        Zhu M, Li Y, Shi J, Feng W, Nie G, Zhao Y. Exosomes as Extrapulmonary Signaling Conveyors for Nanoparticle-Induced Systemic Immune Activation. Small 2012;8:404–12. https://doi.org/10.1002/smll.201101708.

[10]      Tayyaba, Rehman FU, Shaikh S, Tanziela, Semcheddine F, Du T, et al. In situ self-assembled Ag–Fe 3 O 4 nanoclusters in exosomes for cancer diagnosis. Journal of Materials Chemistry B 2020;8:2845–55. https://doi.org/10.1039/C9TB02610J.

[11]      Ferrati S, McConnell KI, Mack AC, Sirisaengtaksin N, Diaz R, Bean AJ, et al. Cellular communication via nanoparticle-transporting biovesicles. Nanomedicine 2014;9:581–92. https://doi.org/10.2217/nnm.13.57.

[12]      Logozzi M, Mizzoni D, Bocca B, di Raimo R, Petrucci F, Caimi S, et al. Human primary macrophages scavenge AuNPs and eliminate it through exosomes. A natural shuttling for nanomaterials. European Journal of Pharmaceutics and Biopharmaceutics 2019;137:23–36. https://doi.org/10.1016/j.ejpb.2019.02.014.

[13]      Marzano M, Bou-Dargham MJ, Cone AS, York S, Helsper S, Grant SC, et al. Biogenesis of Extracellular Vesicles Produced from Human-Stem-Cell-Derived Cortical Spheroids Exposed to Iron Oxides. ACS Biomaterials Science & Engineering 2021;7:1111–22. https://doi.org/10.1021/acsbiomaterials.0c01286.

[14]      Patel M, de Paoli SH, Elhelu OK, Farooq S, Simak J. Cell membrane disintegration and extracellular vesicle release in a model of different size and charge PAMAM dendrimers effects on cultured endothelial cells. Nanotoxicology 2019;13:664–81. https://doi.org/10.1080/17435390.2019.1570373.

[15]      Gurunathan S, Kang M-H, Jeyaraj M, Kim J-H. Palladium Nanoparticle-Induced Oxidative Stress, Endoplasmic Reticulum Stress, Apoptosis, and Immunomodulation Enhance the Biogenesis and Release of Exosome in Human Leukemia Monocytic Cells (THP-1). International Journal of Nanomedicine 2021;Volume 16:2849–77. https://doi.org/10.2147/IJN.S305269.

[16]      Gurunathan S, Kang M-H, Jeyaraj M, Kim J-H. Platinum Nanoparticles Enhance Exosome Release in Human Lung Epithelial Adenocarcinoma Cancer Cells (A549): Oxidative Stress and the Ceramide Pathway are Key Players. International Journal of Nanomedicine 2021;Volume 16:515–38. https://doi.org/10.2147/IJN.S291138.

[17]      Andersson-Willman B, Gehrmann U, Cansu Z, Buerki-Thurnherr T, Krug HF, Gabrielsson S, et al. Effects of subtoxic concentrations of TiO2 and ZnO nanoparticles on human lymphocytes, dendritic cells and exosome production. Toxicology and Applied Pharmacology 2012;264:94–103. https://doi.org/10.1016/j.taap.2012.07.021.

[18]      Hao F, Ku T, Yang X, Liu QS, Zhao X, Faiola F, et al. Gold nanoparticles change small extracellular vesicle attributes of mouse embryonic stem cells. Nanoscale 2020;12:15631–7. https://doi.org/10.1039/D0NR03598J.

[19]      Himly M, Geppert M, Hofer S, Hofstätter N, Horejs‐Höck J, Duschl A. When Would Immunologists Consider a Nanomaterial to be Safe? Recommendations for Planning Studies on Nanosafety. Small 2020;16:1907483. https://doi.org/10.1002/smll.201907483.

Reviewer 4 Report

The paper titled "Nanomaterials exposure, extracellular vesicles biogenesis and adverse cellular outcomes: a scoping review" aims to review and investigate the relevance of EVs in the field of nanotoxicology. The current state of the science on how EVs are modulated by NM exposure and the possible regulation and modulation of signalling pathways and physiological responses.

It is a very interesting and detailed review. The paper is well written and has the makings of a publication. However, some points need attention:

-A small point, keywords in the plural or singular may yield more results.

-The search should be done excluding reviews, otherwise this could lead to an additional "over".

-Can grouping be done by material? What characteristics can be identified from a biological perspective?

Author Response

The authors would like to express their gratitude to the editor and reviewers for considering our manuscript and providing valuable suggestions to improve its quality. A detailed reply to the comments of the reviewers is presented below. The modifications made throughout the text are highlighted in yellow in the new version of the manuscript.

Point 1: A small point, keywords in the plural or singular may yield more results.

Response 1: We agree with the reviewer on this. However, we tested several combinations of keywords at the beginning of our work and we chose to use, mostly, the plural words, since most of the studies found using plural keywords.

Point 2: The search should be done excluding reviews, otherwise this could lead to an additional "over".

Response 2: We agree and thank this reviewer for this observation! Despite increasing the volume of work, the non-exclusion of reviews allows us to carry out a cross-reference finding potential studies to be included in the final review. However, review articles were not included in our studies. We use this method only to cross-references in the search for articles with original data that may not have been found by the search tool. To avoid any problem, we have included two sentences emphasizing this (see line 137 revised manuscript).

Point 3: Can grouping be done by material? What characteristics can be identified from a biological perspective?

Response 3: We thank this reviewer for this comment! In general, we agree that grouping by material would be intuitive for Table 2, however, as we pursued a very stringent and objective selection process for the studies covered in the ScR, and as not the NM per se was the main focus, for instance, several other tables overview in structured way the physicochemical characteristics or models used or initial information relevant for the selection process, we came to the conclusion it was more consistent to stick to the order of the references. Additionally, it has to be noted that we focused on NMs and EVs and that we filtered quite stringently available literature down to a narrow but well-defined list of studies, from this shortlist, however, we were not inclined to draw such general conclusions on the characteristics of NMs on biological outcome. We have included a statement to clarfy this in the revised ScR version, which reads:

Line 304....Notably, no clear trends for a specific type of NM or surface functionalization can be deduced from the limited number of studies that qualified for our ScR (focusing on NMs and EVs). On the biological impact of physicochemical properties of NM in general much has been published during the past decades. From the studies that were selected for this ScR eleven studies....

Round 2

Reviewer 4 Report

The authors have followed my suggestions.